# Central Force Field: Unifying Generative and Discriminative Models While Harmonizing Energy-Based and Score-Based Models

## Abstract

In the pursuit of Artificial General Intelligence, a prevalent approach is to establish a comprehensive unified foundation model that addresses multiple tasks concurrently. However, creating such a model that unifies generative and discriminative models presents significant challenges. This paper aims to realize this unified model aspiration by suggesting the incorporation of a central force field from physics. More precisely, within the framework of this central force field, the potential functions governing the data distribution and the joint data-label distribution become intricately interwoven with a standard discriminative classifier, rendering them well-suited for handling discriminative tasks. Moreover, the central force field exhibits a captivating characteristic: objects located within this field experience an attractive force that propels them towards the center. This phenomenon of centripetal motion, orchestrated by the force field, has the remarkable capability to progressively revert diffused data to its original configuration, thereby facilitating the execution of generative tasks. Our proposed method adeptly bridges the realms of energy-based and score-based models. Extensive experimental validation attests to the effectiveness of our approach, showcasing not only its prowess in image generation benchmarks but also its promising competitiveness in image classification benchmarks.

## 1 Introduction

In recent years, significant progress has been made in multi-task large models, offering a glimpse of the potential for Artificial General Intelligence (AGI) [1]. These large models, often referred to as foundation models (Bommasani et al., 2021), are expected to serve as versatile intelligent agents capable of handling various general tasks (Di Palo et al., 2023). Examples of such models include ChatGPT (OpenAI, 2022) and GPT4 (OpenAI, 2023), which excel at addressing diverse language tasks, as well as INTERN (Shao et al., 2021), specialized in discriminative tasks, and Gato (Reed et al., 2022) and GITM (Zhu et al., 2023), proficient in various gaming tasks.

While the idea of unified foundation models is appealing, the task of creating a cohesive generative and discriminative foundation model is indeed quite challenging. Some generative and discriminative hybrids (Lasserre et al., 2006) have been primarily designed for classification tasks and may not be suitable for data generation tasks [2]. Further, some arts attempted to blend generative and discriminative tasks (Tu, 2007; Grathwohl et al., 2019; Santurkar et al., 2019; Wang & Torr, 2022; Lee et al., 2018; Jin et al., 2017; Xie et al., 2016; Du & Mordatch, 2019). These methods have skillfully integrated insightful techniques, including bonus learning (where discriminative and generative learning mutually benefit from each other) (Tu, 2007; Lee et al., 2018; Jin et al., 2017), energy-based models (Grathwohl et al., 2019; Xie et al., 2016; Du & Mordatch, 2019), adversarial training models (Santurkar et al., 2019), and novel sampling (Wang & Torr, 2022). However, due to their constrained capacity to efficiently model high-dimensional data spaces, these arts face

---

[1]It is important to approach this topic with caution and humility, as it is possible that large models may not represent the ultimate solution for AGI. However, it is undeniable that these foundation models play a pivotal role in driving AGI development.

[2]Note that generative models do not necessarily imply proficiency in generation tasks.

challenges in achieving optimal results in image generation tasks, typically also experiencing issues related to learning instability and inefficiency.

In this paper, we aim to pursue a foundational model that unifies generative and discriminative models. To accomplish this, we draw inspiration from Physics. Many of the most fundamental forces in the universe, such as gravity (as described by Newton's law (Newton, 1687)), electrostatic force (as described by Coulomb's law (Coulomb, 1785)), and elasticity (as described by Hooke's law (Hooke)), are central forces (Taylor & Taylor, 2005), which have the potential to contribute to achieving this pursuit. Precisely, a central force field $\mathbf{F}$ is a gradient field of a potential function $E$, i.e.: $\mathbf{F} = \frac{\partial E}{\partial \mathbf{x}}$ (see a simple example in Fig. 1). We can leverage the potential function $E$ to model discriminative tasks: In a central force field, the potential functions for the data distribution $p(\mathbf{x})$

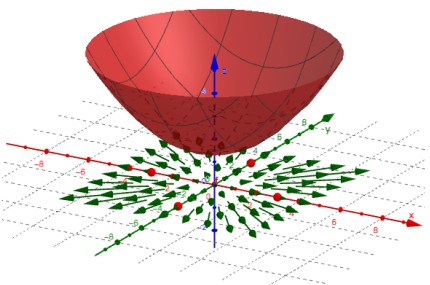

Figure 1: Example of a center force field, where the arrow points in the opposite direction of the force, indicating that the central force is directed towards the center.

and the joint data-label distribution $p(\mathbf{x}, y)$ are intricately connected to a standard discriminative classifier of $p(y|\mathbf{x})$, rendering the potential function suitable for modeling discriminative tasks. Simultaneously, we can employ the central force ($\mathbf{F}$) to model generative tasks: A central force field possesses a truly intriguing property, whereby objects at any point in this field are compelled to move towards the center by an attractive force impressed upon them, thus facilitating the accomplishment of generative tasks.

Moreover, our method can reconcile energy-based models (LeCun et al., 2006; Teh et al., 2003; LeCun & Huang, 2005) and score-based models (Sohl-Dickstein et al., 2015; Song et al., 2020). It aligns with energy-based models in employing potential functions, while differentiating by explicitly modeling force fields. The innovation lies in our approach's direct utilization of force fields for training, effectively circumventing the training hurdles often encountered in traditional energy-based models. Notably, we have refined LeCun et al. (2006)'s energy definition to ensure compatibility with discriminative models. This adjustment addresses a specific issue with the original definition where the lowest-energy point might be inaccessible. Regarding score-based models (e.g., diffusion models (Ho et al., 2020; Nichol & Dhariwal, 2021; Dhariwal & Nichol, 2021)), our method's affinity is discernible through the intrinsic diffusion and denoising mechanisms. The primary distinction between score-based models and our approach lies in the conceptually different understanding of gradients. Specifically, score-based approaches require their model (e.g., a U-Net) to output the derivatives of the data density function, rather than the data density function itself. In contrast, our approach uses a network to output the data density function / potential function. Consequently, derivatives of the data density function can naturally be the gradients of a network, which exactly aligns with the gradient computation in the backpropagation of deep learning. This alignment makes our method adhere more closely to physical principles.

We conducted comprehensive experiments to validate the effectiveness of our method in both image classification and generation tasks. The results reveal that our method not only achieves remarkably favorable results in image generation benchmarks but also demonstrates strong competitiveness in image classification benchmarks.

In summary, this paper makes three contributions: **Firstly**, we draw inspiration from physics and propose a foundation model that unifies generative and discriminative models using the central force field. The fascinating property of the central force field enables the attainability of this unified model because the potential function in a central force field can be employed for modeling discriminative tasks, while the central force can be utilized for modeling generative tasks. **Secondly**, we provide a clear and comprehensive explanation of the connections between our method and two mainstream prior arts, namely energy-based models and score-based models. We highlight the distinctions between our approach and these arts. Our analytical results demonstrate that our method effectively harmonizes both energy-based and score-based models. **Thirdly**, we present extensive experimental results that showcase the remarkable effectiveness of our method. Our approach not only achieves highly favorable results in image generation benchmarks but also demonstrates strong competitiveness in image classification benchmarks, along with adversarial robustness tasks.

## 2 RELATED WORK

**Hybrid Discriminative and Generative Models.** The pursuit of unified foundation models, capable of integrating generative and discriminative functionalities, has been an enticing research direction in recent years. Notably, certain models have primarily been designed for classification tasks and may not readily translate to data generation tasks (Lasserre et al., 2006). Several studies have ventured into the integration of generative and discriminative tasks, seeking to harness the synergistic potential of both components. These endeavors have yielded promising results and have introduced innovative techniques into the field. Noteworthy among these techniques is "bonus learning (Tu, 2007; Lee et al., 2018; Jin et al., 2017)," a paradigm where discriminative and generative models mutually enhance each other's performance. Additionally, energy-based models (Grathwohl et al., 2019; Xie et al., 2016; Du & Mordatch, 2019), adversarial training models (Athalye et al., 2018; Santurkar et al., 2019; Engstrom et al., 2019; Tsipras et al., 2018; Schott et al., 2018), and insightful sampling strategies (Wang & Torr, 2022) have been incorporated into these unified models to improve their efficacy. Despite the ingenuity in integrating generative and discriminative tasks, unified foundation models face challenges when applied to image generation tasks. The inherent limitation in efficiently modeling high-dimensional data spaces poses significant obstacles. These challenges manifest in difficulties achieving optimal results in image generation tasks, often accompanied by issues related to learning instability and inefficiency. In conclusion, while the pursuit of unified foundation models holds promise for advancing the field, it is essential to recognize the intricate challenges involved. Understanding the nuances and complexities of creating cohesive generative and discriminative models is crucial for guiding future research efforts in this evolving domain.

**Energy-Based and Score-Based Models.** Score-based generative models (Sohl-Dickstein et al., 2015; Song et al., 2020; Ho et al., 2020; Nichol & Dhariwal, 2021; Dhariwal & Nichol, 2021), often referred to as "likelihood-free" models, estimate the gradient of the log-likelihood with respect to the data. These models have gained popularity due to their flexibility in handling complex data distributions and their ability to generate high-quality samples. Energy-based models (EBMs) are a class of models that associate a scalar energy score with each data sample (LeCun et al., 2006; Teh et al., 2003; LeCun & Huang, 2005; Nijkamp et al., 2019). Lower energy scores are assigned to more plausible data points, making EBMs capable of generating samples by finding configurations with minimal energy. Notable examples of EBMs include Restricted Boltzmann Machines (RBMs) (Nair & Hinton, 2010; Ackley et al., 1985; Hinton, 2012) and Deep Boltzmann Machines (DBMs) (Hinton et al., 2006; Salakhutdinov & Hinton, 2009). Score-based and energy-based models are distinct in generative modeling, with unique features. Combining elements of both has gained interest. Schroder et al. (Schröder et al., 2023) propose a hybrid approach, using score-based models to improve training stability and generative capabilities for energy-based models. Chao et al. (Chao et al., 2023) incorporate score-based models as prior distributions in energy-based models to capture complex data distributions. Nonetheless, the process of integrating energy-based and score-based models still requires significant work. Furthermore, Section 3.4 highlights the key distinctions between our approach and both energy-based and score-based models.

**Central Force Field.** Central forces, as described by Newton's law (gravity, (Newton, 1687)), Coulomb's law (electrostatic force, (Coulomb, 1785)), and Hooke's law (elasticity, (Hooke)), constitute many of the fundamental forces in the universe (Taylor & Taylor, 2005).

## 3 METHODOLOGY

In this section, we present our methodology in detail. To begin with, we introduce the physical meaning of the central force field (Section 3.1). Subsequently, we demonstrate how we utilize the potential function within a central force field to formulate discriminative models (Sectioin 3.2). Moving forward, we elaborate on how the central force can be effectively used for formulating generative models (Section 3.3). Finally, we provide a clear explanation of the connection between our method and energy-based models as well as score-based models (Section 3.4).

### 3.1 INTRODUCTION TO CENTRAL FORCE FIELD

In the universe, many of the most fundamental forces, such as gravity (as described by Newton's law (Newton, 1687)), electrostatic force (as described by Coulomb's law (Coulomb, 1785)), and

elasticity (as described by Hooke's law (Hooke)), are central forces (Taylor & Taylor, 2005). In classical mechanics, a central force is a force directed from an object towards a fixed center point, with magnitude determined solely by the distance between the object and the center point:

$$\frac{\mathbf{F}}{\|\mathbf{F}\|} = \frac{\mathbf{x}}{\|\mathbf{x}\|},$$

(1)

where $\mathbf{F}$ denotes a central force and $\mathbf{x}$ is a vector pointing from the object towards the center. Moreover, the forces in a field can always be mathematically expressed as gradient fields, wherein they correspond to the gradient of a potential energy function denoted by $E(\mathbf{x})$, i.e.:

$$\mathbf{F} = \frac{\partial E(\mathbf{x})}{\partial \mathbf{x}}.$$

(2)

The study of central force fields holds significance in classical mechanics, where a specific category of problems is devoted to them, known as *central-force problems*. The central-force problem pertains to the investigation of a particle's motion within a central potential field. Solving this problem holds great significance in classical mechanics for two primary reasons: Firstly, numerous naturally occurring forces are central forces. Secondly, certain intricate problems in classical physics, like the two-body problem, can be effectively reduced to a central-force problem.

## 3.2 POTENTIAL FUNCTION FOR MODELING DISCRIMINATIVE MODELS

In the realm of machine learning, conventionally, each individual data point denoted as $\mathbf{x}$, along with its corresponding label $y$, is typically represented by a scalar value, a configuration that is often modeled using a neural network with parameters $\theta$. Herein, the notation $g_\theta(\mathbf{x}, y)$ is employed to denote this scalar entity. With this function $g$, we are able to articulate the distribution $p(\mathbf{x}, y)$:

$$p(\mathbf{x}, y) = \frac{g_\theta(\mathbf{x}, y)}{\int_\mathbf{x} \int_y g_\theta(\mathbf{x}, y) dy d\mathbf{x}}, \quad p(\mathbf{x}) = \frac{\int_y g_\theta(\mathbf{x}, y) dy}{\int_\mathbf{x} \int_y g_\theta(\mathbf{x}, y) dy d\mathbf{x}}.$$

(3)

.

Dating back to Lecun et al.'s definition (LeCun et al., 2006), prior arts (Grathwohl et al., 2019; Du & Mordatch, 2019) typically established a connection between a probability function $p(\mathbf{x})$ and an energy function $E(\mathbf{x})$ using the following definition:

$$\text{LeCun et al. (2006)'s energy:} \quad p(\mathbf{x}, y) = \frac{e^{-E(\mathbf{x}, y)}}{\int_\mathbf{x} \int_y e^{-E(\mathbf{x}, y)} dy d\mathbf{x}}, \quad p(\mathbf{x}) = \frac{e^{-E(\mathbf{x})}}{\int_\mathbf{x} e^{-E(\mathbf{x})} d\mathbf{x}}.$$

(4)

We have encountered an issue with this definition: the lowest-energy point might be inaccessible (please see Appendix for the principle insight and Sec. 4.4 for the empirical analysis). Therefore, we propose a refined definition:

$$\text{Our energy:} \quad p(\mathbf{x}, y) = \frac{e^{-\log E(\mathbf{x}, y)}}{\int_\mathbf{x} \int_y e^{-\log E(\mathbf{x}, y)} dy d\mathbf{x}}, \quad p(\mathbf{x}) = \frac{e^{-\log E(\mathbf{x})}}{\int_\mathbf{x} e^{-\log E(\mathbf{x})} d\mathbf{x}}.$$

(5)

By amalgamating Eqn. 5 with Eqn. 3, the formulation of the energy function can be acquired:

$$e^{-\log E(\mathbf{x}, y)} = g_\theta(\mathbf{x}, y), \quad e^{-\log E(\mathbf{x})} = \int_y g_\theta(\mathbf{x}, y) dy,$$

(6)

which yields:

$$E(\mathbf{x}, y) = e^{-\log g_\theta(\mathbf{x}, y)}, \quad E(\mathbf{x}) = e^{-\log \int_y g_\theta(\mathbf{x}, y) dy}.$$

(7)

In the conventional context, the definition of $g_\theta(\mathbf{x}, y)$ takes the form of:

$$g_\theta(\mathbf{x}, y) = e^{f_\theta(\mathbf{x})[y]}.$$

(8)

By integrating Eqn. 8 with 7, the conclusive form of the energy function can be readily deduced:

$$E(\mathbf{x}, y) = e^{-\log e^{f_\theta(\mathbf{x})[y]}} = e^{-f_\theta(\mathbf{x})[y]}, \quad E(\mathbf{x}) = e^{-\log \sum_y e^{f_\theta(\mathbf{x})[y]}}.$$

(9)

Based on the two formulations presented in Eqn. 3, the derivation of $p(y|\mathbf{x})$ can be acquired:

$$p(y|\mathbf{x}) = \frac{p(\mathbf{x}, y)}{p(\mathbf{x})} = \frac{g_\theta(\mathbf{x}, y)}{\int_y g_\theta(\mathbf{x}, y) dy}. \tag{10}$$

By incorporating Eqn. 8 into Eqn. 10, a conventional expression for $p(y|\mathbf{x})$ emerges:

$$p(y|\mathbf{x}) = \frac{e^{f_\theta(\mathbf{x})[y]}}{\sum_{y'} e^{f_\theta(\mathbf{x})[y']}}, \tag{11}$$

signifying a classical classifier within a discriminative model. Eqn. 11 precisely represents the pre-eminent expression employed in conventional neural network classifiers, underscoring the inherent correlation between an energy function and a classification function.

In summary, both the potential energy function and the discriminative model utilize a shared neural network denoted as $f_\theta$. Consequently, the potential function can be effectively employed in the modeling of discriminative models. Additional comparisons with prior energy-based models, including JEM (Grathwohl et al., 2019), are detailed in Section 3.4.

### 3.3 CENTRAL FORCE FOR MODELING GENERATIVE MODELS

Upon establishing the definition of the potential function in Eqn. 9, the conventional line of thought might lead one to consider utilizing energy-based models for the purpose of sample generation, which may entail inherent limitations as mentioned in Section 3.4. Here, we present a distinct perspective that enables us to fully capitalize on the inherent strengths of a central force field.

Let $\mathbf{x}_0$ denote a center, then Eqn. 1 becomes:

$$\frac{\mathbf{x}_t - \mathbf{x}_0}{\|\mathbf{x}_t - \mathbf{x}_0\|} = \frac{\mathbf{F}(\mathbf{x}_t)}{\|\mathbf{F}(\mathbf{x}_t)\|}, \tag{12}$$

where $\mathbf{F}(\mathbf{x}_t)$ is computed by:

$$\mathbf{F}(\mathbf{x}_t) = \frac{\partial E(\mathbf{x}_t)}{\partial \mathbf{x}_t}. \tag{13}$$

Let $\frac{\|\mathbf{x}_t - \mathbf{x}_0\|}{\sqrt{d}} = \lambda_t$ where $d$ is a constant denoting the dimension of the data. We can derive the subsequent expression from Eqn. 12:

$$\mathbf{x}_t = \mathbf{x}_0 + \lambda_t \sqrt{d} \frac{\mathbf{F}(\mathbf{x}_t)}{\|\mathbf{F}(\mathbf{x}_t)\|}. \tag{14}$$

Eqn. 14 indicates that when an object is moved from $\mathbf{x}_0$ to $\mathbf{x}_t$, it experiences a central force. Without loss of generality, the displacement from $\mathbf{x}_0$ to $\mathbf{x}_t$ can be delineated by $\epsilon$:

$$\mathbf{x}_t = \mathbf{x}_0 + \frac{\sqrt{1 - \bar{\alpha}_t}}{\sqrt{\bar{\alpha}_t}} \epsilon, \tag{15}$$

where $\{\bar{\alpha}_t\} \in (0, 1]$ are a set of predefined constants adhering to specific monotonically increasing functions with regard to time $t$. Let $\lambda_t = \frac{\sqrt{1 - \bar{\alpha}_t}}{\sqrt{\bar{\alpha}_t}}$. Upon comparing Eqn. 14 and Eqn. 15, we can derive the following relationship:

$$\sqrt{d} \frac{\mathbf{F}(\mathbf{x}_t)}{\|\mathbf{F}(\mathbf{x}_t)\|} = \epsilon. \tag{16}$$

The left side of Eqn. 16 signifies a force with a magnitude of $\sqrt{d}$ and an orientation of $\frac{\mathbf{F}(\mathbf{x}_t)}{\|\mathbf{F}(\mathbf{x}_t)\|}$. The right side of Eqn. 16 pertains to the displacement through which the object deviates from the central point. More precisely, given an object in an initial position $\mathbf{x}_t$ (e.g., a Gaussian noise image), it is feasible to systematically shift it towards $\mathbf{x}_0$ (e.g., a natural image) in a gradual manner. This process involves the computation of the force $\mathbf{F}(\mathbf{x}_t)$ upon the object through Eqn. 13, followed by the application of Eqn. 16 to determine the displacement $\epsilon$. Finally, the object can be re-positioned towards the central point by utilizing the calculated displacement, thus achieving sample generation.

### 3.4 Harmonizing Energy-Based and Score-Based Model

Our model adeptly reconciles the principles of energy-based and score-based models. The interrelation and differentiation of our model from these paradigms are expounded as follows.

*Association with Energy-Based Models.* This association becomes evident due to the shared utilization of the potential function concept ($E(\mathbf{x})$) in both models.

*Distinction from Energy-Based Models.* The distinctions are three-fold.

- **Explicit Formulation of Force Field:** Our approach explicitly formulates the force field, a concept that represents the gradient of energy in the physical realm but is surprisingly often absent within the energy-based model framework.

- **Refinement of Lecun et al.'s Energy Definition:** We have refined Lecun et al.'s energy definition to ensure compatibility with discriminative models. This adjustment addresses a specific issue with the original definition where the lowest-energy point might be inaccessible.

- **Direct Training vs. Normalization Challenges:** Energy-based models encounter a challenge in directly determining the precise likelihood of a point due to the presence of an unknown normalization constant (analogous to the denominator in Eqn. 3). As a consequence, training energy-based models necessitates the adoption of techniques such as contrastive divergence (LeCun et al., 2006; Lippe, 2022; Grathwohl et al., 2019) that is similar to adversarial training (Madry et al., 2017). It's worth noting that, although contrastive divergence is used to train energy-based models, JEM (Grathwohl et al., 2019) explicitly acknowledges this as a major limitation in its paper, as well as its counterpart paper (Du & Mordatch, 2019). They admit that training can be highly unstable and prone to divergence, despite employing numerous regularization techniques to mitigate this instability. In contrast, our method, thanks to its force field modeling, is capable of direct training, avoiding the challenges associated with normalization constant issues and instability during training.

This tripartite distinction underlines the unique attributes that set our model apart from traditional energy-based approaches.

*Association with Score-Based Models.* The connection of our approach to a score-based model is readily apparent. To elucidate, Eqn. 15 can be elegantly reformulated as follows:

$$\mathbf{x}_t = \sqrt{\bar{\alpha}_t}\mathbf{x}_0 + \sqrt{1 - \bar{\alpha}_t}\epsilon, \tag{17}$$

which coherently aligns with the diffusion process in a score-based model. When it comes to the denoising process, a score-based model employs a network ($\epsilon_\theta$) to predict and subsequently reduce noise. This process shares similarities with the centripetal motion mechanism used in our approach:

$$\mathbf{x}_0 = \mathbf{x}_t - \frac{\sqrt{1 - \bar{\alpha}_t}}{\sqrt{\bar{\alpha}_t}}\epsilon_\theta \text{ (for score-based)}, , \mathbf{x}_0 = \mathbf{x}_t - \lambda_t\sqrt{d}\frac{\mathbf{F}(\mathbf{x}_t)}{\|\mathbf{F}(\mathbf{x}_t)\|} \text{ (for ours) }. \tag{18}$$

*Distinction from Score-Based Models.* The differences manifest as follows:

- **Conceptually Different Understanding of Gradients:** There are many significant differences between score-based models and our method, but we would like to highlight the most significant one here, which is the conceptually different definition of gradients. Specifically, score-based approaches require their model (e.g., a U-Net $\epsilon_\theta$) to output the derivatives of the data density function, rather than the data density function itself. In contrast, our approach uses a network to output the data density function or the potential function. Consequently, derivatives of the data density function can naturally be the gradients of a network, which exactly aligns with the gradient computation in the backpropagation of deep learning. This alignment makes our method adhere more closely to physical principles.

## 4 Experiments

Given our method's capacity to unify generative and discriminative models, we primarily present experimental results on datasets empowered with classification annotations, including CIFAR-10

Table 1: CIFAR-10 hybrid modeling results.
*Residual Flow (Chen et al., 2019), Glow (Kingma & Dhariwal, 2018), and IGEBM (Du & Mordatch, 2019), JEM (Grathwohl et al., 2019).*

| Model | Acc.% ↑ | IS↑ | FID↓ ↓ | Stability ↑ | Inference steps |
|---|---|---|---|---|---|
| Residual Flow | 70.3 | 3.6 | 46.4 | N/A | N/A |
| Glow | 67.6 | 3.92 | 48.9 | N/A | N/A |
| IGEBM | 49.1 | 8.3 | 37.9 | N/A | N/A |
| JEM $p(x\|y)$ factored | 30.1 | 6.36 | 61.8 | N/A | N/A |
| JEM | 92.9 | 8.76 | 38.4 | 50% | > 1000 |
| **Central Force Field (Eqn. 4)** | N/A | N/A | N/A | 0% | N/A |
| **Central Force Field (Eqn. 5)** | **93.6** | **9.01** | **24.9** | **100%** | **50** |

(Krizhevsky et al., 2009), SVHN (Netzer et al., 2011), and CIFAR100 (Krizhevsky et al., 2009). We also conduct unconditional image synthesis on the CelebA dataset (Liu et al., 2015).

We employ a Wide Residual Network as described in (Grathwohl et al., 2019) for these four dataset. This architecture is initially proposed by (Zagoruyko & Komodakis, 2016) and is modified by (Grathwohl et al., 2019) to enhance stability. To ensure a fair comparison, we borrow the training hyper-parameters from (Grathwohl et al., 2019).

In our evaluation of discriminative tasks, we utilize classification accuracies as the primary evaluation metric. For the generative tasks, we employ three metrics: Inception Scores (IS) (Salimans et al., 2016), Frechet Inception Distance (FID) (Heusel et al., 2017), and qualitative visualization.

## 4.1 COMPARISONS WITH HYBRID MODELS

We began by evaluating the effectiveness of our approach in hybrid modeling tasks involving both discriminative and generative objectives.

To gauge the performance of our approach, we conducted a comprehensive comparative analysis using the CIFAR-10 dataset, comparing it against prior arts. Multiple metrics, including classification accuracy and generation quality scores, were taken into account. The results presented in Table 1 and Fig. 3 indicate that our method surpasses prior hybrid arts, excelling the joint discriminative and generative tasks. Previous energy-based models have encountered challenges in accurately computing the likelihood of a data point because of the unknown normalization denominator in Eqn. 3. To address this issue, they have taken one of two routes: either they neglect this term, albeit at the expense of theoretical accuracy in discriminative models (Du & Mordatch, 2019; Xie et al., 2016), or they employ a contrastive divergence technique (LeCun et al., 2006; Lippe, 2022; Grathwohl et al., 2019), akin to adversarial training (Madry et al., 2017), as a surrogate for optimization (Grathwohl et al., 2019). The former approach renders these models less competitive in discriminative tasks (e.g., 49.1% for IGEBM in Table 1), while the latter results in lower learning instability (see "Stability") and inferior performance compared to our method (see JEM in Table 1).

It's noteworthy that prior hybrid methods have seldom reported results on the CIFAR-100 and SVHN datasets, except that JEM provides qualitative visualization results on these two datasets, albeit lacking numerical quantitative results. Hence, we performed a qualitative comparison of our method with JEM, focusing on visualization. As demonstrated in Fig. 3, our approach look more promising than JEM. Besides, we also provide quantitative results for our method in Table 2.

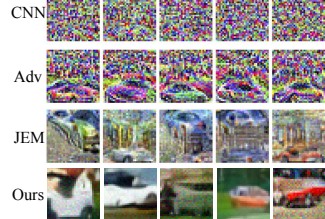

**Stability.** JEM (Grathwohl et al., 2019) noted the potential instability in training energy-based models, particularly for hybrid classification and generation tasks. They conducted experiments involving

Figure 2: Feature visualization.

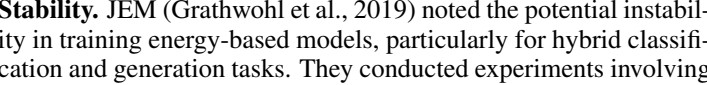

various regularization techniques to address this issue but faced challenges in finding suitable regularization methods that could stabilize learning without adversely affecting the performance of both classification and generation tasks. In our attempts to replicate JEM's experiments, we indeed observed training instability. Even when using their carefully tuned hyperparameters, our experiments reproduced their results ten times, with only 5 instances being successful. Conversely, when we ran our code for ten trials, all of them were successful (refer to Table 1). This demonstrates the stability of our approach.

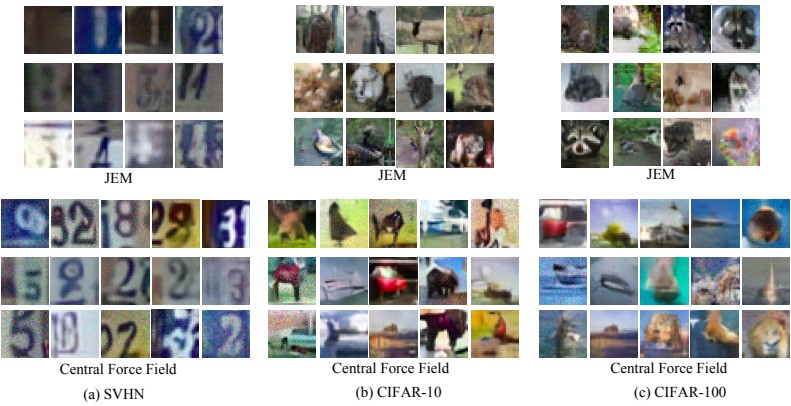

Figure 3: Results of conditional image synthesis on SVHN, CIFAR-10, and CIFAR-100.

Another significant metric is the model's confidence in classifying nonsensical input. To assess this metric, we conduct the following experiments: starting with Gaussian noise $\mathbf{x}$, we maximize $p(y|\mathbf{x})$ to determine if the optimized $\mathbf{x}$ visually resembles an image associated with label $y$ (Athalye et al., 2018; Santurkar et al., 2019; Engstrom et al., 2019; Tsipras et al., 2018; Schott et al., 2018). As depicted in Fig. 2, energy-based models such as JEM and adversarially trained models can generate samples that align with human perception, whereas vanilla classifiers cannot. It's important to note that when directly maximizing $p(y|\mathbf{x})$, JEM cannot produce samples as meaningful as those generated through its usual sampling process; the produced samples appear significantly inferior to those generated by our method. This highlights a weakness in JEM. In fact, to generate meaningful samples, JEM relies on starting with "half-ready" samples stored in a buffer, which introduces an element of complexity and reduces its elegance. In contrast, our method can directly generate promising samples, showcasing the superiority of our approach.

## 4.2 COMPARISONS WITH DISCRIMINATIVE MODELS

Further, we conduct a comparative analysis of our method alongside discriminative models. To ensure a fair assessment, we require that all competing methods adopt identical neural architectures to ours, thereby minimizing any variations caused by architectural dif-

Table 2: CIFAR-10 discriminative results. *WideResNet(Zagoruyko & Komodakis, 2016).*

|  | CIFAR-10 | CIFAR-100 | SVHN |
|---|---|---|---|
| Wide ResNet | 95.8% | 79.5% | 97.7% |
| **Central Force Field** | 93.6% | 73.7% | 95.4% |

ferences. We do not delve into the exploration of alternative architectures, as the primary focus of this paper is not architecture design. For the CIFAR-10, CIFAR-100, and SVHN datasets, we have opted to use Wide Residual Networks, as they are widely acknowledged as state-of-the-art models in these contexts. The results presented in Table 2 demonstrate that our method achieves comparable accuracies when compared to state-of-the-art discriminative models.

## 4.3 COMPARISON WITH GENERATIVE MODELS

### 4.3.1 CONDITIONAL IMAGE SYNTHESIS

Given our focus on training a hybrid model, our model naturally belongs to conditional generative models, as it leverages label information. Therefore, we will primarily evaluate the performance of our method in conditional image synthesis tasks, with a specific focus on the CIFAR-10 dataset. We conducted a comparison of our model with state-of-the-art approaches in conditional image synthesis tasks using the CIFAR-10 dataset. The results are presented in Table 3. It is evident from the table that our model achieves competitive performance when compared to leading conventional conditional generative models, such as GANs and diffusion models.

Beyond CIFAR-10, we also conduct conditional image synthesis on SVHN and CIFAR-100 (see Fig. 3), which demonstrates the promising results of our method.

### 4.3.2 UNCONDITIONAL IMAGE SYNTHESIS

Our method can also adapt to unconditional generative tasks by exclusively considering the term $E(\mathbf{x})$ while omitting $E(\mathbf{x}, y)$. To further illustrate this adaptability, we conducted additional unconditional generation on CIFAR-10, CIFAR-100, SVHN, and CelabA.

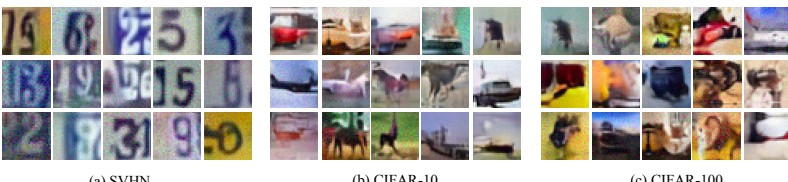

|          |            |             |
|----------|------------|-------------|
| (a) SVHN | (b) CIFAR-10 | (c) CIFAR-100 |

Figure 4: Results of unconditional image synthesis on SVHN, CIFAR-10, and CIFAR-100.

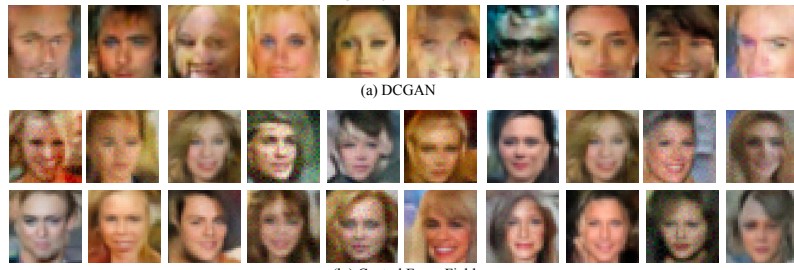

(a) DCGAN

(b ) Central Force Field

Figure 5: Results of unconditional image synthesis on CelebA 32×32.

We extended our comparative analysis to evaluate our method against state-of-the-art approaches in this task using CIFAR-10. The results, detailed in Table 3, reveal that our method achieves promising performance. Additionally, we present the results for the CelebA dataset in Fig. 4 and Fig. 5, which further demonstrate the promising performance achieved by our method. Specifically, on CelebA, our method shows comparable performance with DCGAN Radford et al. (2015).

### 4.4 ABLATION STUDY

Unlike previous energy-based models that require extensive hyperparameter tuning, involving techniques like small learning rates, additional restarting, and numerous SGLD steps, along with the addition of essential modules such as buffers for storing "half-ready" samples for initialization, our method is streamlined. It comprises only a gradient field loss and a classification component, making it simpler and more efficient. Consequently, conducting an ablation study on our method is straightforward. One aspect we need to investigate in our ablation study is our proposed energy definition in Eqn. 5, which differs from Lecun et al.'s definition in Eqn. 4. To explore this, we substitute our energy with Eqn. 4 and rerun the experiment on CIFAR-10. The results are presented in Table 1, indicating that our new definition enhances learning stability. A detailed analysis is provided in the appendix.

### 5 CONCLUSION

In summary, this paper integrate a central force field from physics. Within this framework, the interconnection of potential functions governing data and joint data-label distributions with a standard discriminative classifier enables effective handling of discriminative tasks. Moreover, the central force field exhibits an attractive force, facilitating the progressive reversion of diffused data to its original configuration for generative tasks. Our method successfully bridges energy-based and score-based models, as demonstrated through extensive experiments, highlighting its efficacy in image generation benchmarks and its promising competitiveness in image classification benchmarks.

Table 3: CIFAR10 results. *EBM (Du & Mordatch, 2019), JEM (Grathwohl et al., 2019), BigGAN(Brock et al., 2018), StyleGAN2+ADA (v1) (Karras et al., 2020), Diffusion (original) (Sohl-Dickstein et al., 2015), Gated PixelCNN (Van den Oord et al., 2016), Sparse Transformer (Child et al., 2019), PixelIQN (Ostrovski et al., 2018), EBM (Du & Mordatch, 2019), NCSNv2 (Song & Ermon, 2020), NCSN (Song & Ermon, 2019), SNGAN (Miyato et al., 2018), SNGAN-DDLS (Che et al., 2020), StyleGAN2+ADA (v1) (Karras et al., 2020).*

| Model | IS | FID |
|-------|-----|-----|
| **Conditional** | | |
| EBM | 8.30 | 37.9 |
| JEM | 8.76 | 38.4 |
| BigGAN | 9.22 | 14.73 |
| StyleGAN2+ADA(v1) | **10.06** | **2.67** |
| Central Force Field | 9.01 | 24.9 |
| **Unconditional** | | |
| Diffusion (original) | | |
| Gated PixelCNN | 4.60 | 65.93 |
| Sparse Transformer | | |
| PixelIQN | 5.29 | 49.46 |
| EBM | 6.78 | 38.2 |
| NCSNv2 | | 31.75 |
| NCSN | 8.87±0.12 | 25.32 |
| SNGAN | 8.22±0.05 | 21.7 |
| SNGAN-DDLS | 9.09±0.10 | 15.42 |
| StyleGAN2+ADA(v1) | **9.74** ± 0.05 | 3.26 |
| DDPM ($L_{\text{simple}}$) | 9.46±0.11 | **3.17** |
| **Central Force Field** | 8.53±0.18 | 32.68 |
| JEM | 7.79 | |
| **Central Force Field** | 8.53±0.18 | 32.68 |

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

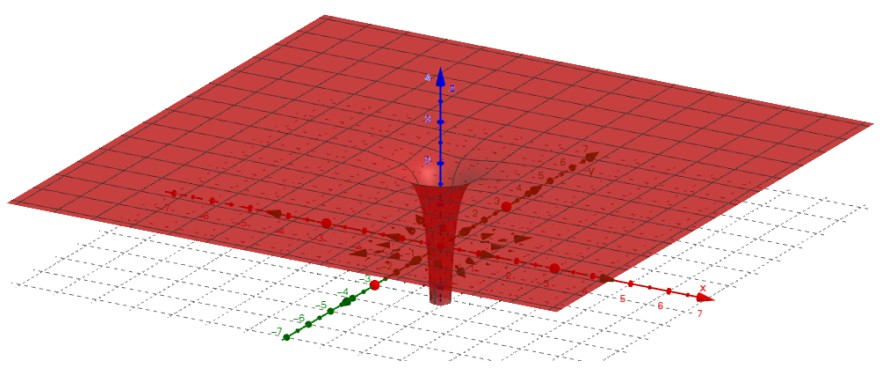

Figure 6: Example of a force field where the lowest-energy point might be inaccessible.

## A  ANALYSIS OF THE REFINED DEFINITION OF THE ENERGY (EQN. 5)

If we do not refine the energy definition from LeCun et al. (2006), Eqn. 9 will take the following form:

$$E(\mathbf{x}, y) = -\log e^{f_\theta(\mathbf{x})[y]} = -f_\theta(\mathbf{x})[y], \quad E(\mathbf{x}) = -\log \sum_y e^{f_\theta(\mathbf{x})[y]}. \tag{19}$$

In energy-based models, generating a sample involves minimizing the energy. Ideally, we prefer an energy function resembling Fig. 1, where the lowest-energy point is easily accessible.

However, considering the classifier in Eqn. 11, we observe that $f_\theta(\mathbf{x})[y]$ represents the pre-softmax logits for a specific class. In discriminative tasks, we aim for $f_\theta(\mathbf{x})[y]$ to be as large as possible. This requirement could pose a problem for Eqn. 19 because it implies the lowest-energy point might be inaccessible, linearly approaching $-\infty$. An example of such a central force field is illustrated in Fig. 6. With this kind of force field, convergence to an optimally low energy is unachievable—wherever you reach, there is always another point with even lower energy. This unbounded nature of the optimization leads to practical issues, often causing numerical instability during training.

With our refined definition of energy, the convergence of Eqn. 9 becomes more manageable as the energy converges to zero. By setting a sufficiently small constant $c$ as the convergence condition, we can easily achieve this condition by maximizing the pre-softmax logits $f_\theta(\mathbf{x})[y]$.

## B  COMPUTATIONAL COST

Similar to DDIM, our method can utilize as few as 50 steps for sampling during the inference stage, while in JEM, a minimum of 1,000 SGLD steps is typically required (as indicated in Table 1), and their recommendation is to use as many steps as possible. This comparison validates the efficiency of our method.

