# OpenReview forum: "Central Force Field: Unifying Generative and Discriminative Models While Harmonizing Energy-Based and Score-Based Models"
_ICLR.cc/2024/Conference — ICLR 2024 Conference Withdrawn Submission_

### Official Review · Reviewer_3hco · 2023-10-31

**Soundness:** 2 fair
**Presentation:** 2 fair
**Contribution:** 2 fair
**Rating:** 3
**Confidence:** 4

**Summary:**

This paper has proposed a framework for unified foundation models with ideas inspired by the central force field. The proposed method could unify both generative and discriminative models by directly modeling the data density function or potential function by using a neural network.

**Strengths:**

This paper integrates the idea of the central force field from physics in a unified framework for discrimination and generation. The proposed model could facilitate the progressive reversion of the diffusion model by modeling the "attractive force" and has bridged the gap between energy-based and score-based models.

**Weaknesses:**

Although this paper proposes a unified framework as the foundation for both generative and discriminative models, I can't see a significant difference between the proposed model and the existing energy-based model. Comparing Eq(4) and Eq(5), the only change is the modeling of the energy function, where the proposed model is expected to output a positive value due to the log function, and the author doesn't give a clear explanation about the motivation behind this. In fact, a discussion about the connection between these two models, including the descriptive model, could also be found as early as in [1][2][3], which is missed in the related work. Furthermore, the experiment results in Table 2 and Table 3 are not very impressive, especially compared with recent energy-based models or score-based models like [4][5]. Additionally, there are some recent and important EBM baseline methods missing in Table 3. For example, CoopFlow [6], VAEBM [7], CoopNet [8], CoopVAEBM [9], FEC [10] are important energy-based generative models that the author might consider to compare with.

Except for this, I also have some comments for the bad organization:
1. Select the same number of images in Figure 3 for JEM and the Central Force Field.
2. Keep the same precision in Table 3.

[1] A tale of three probabilistic families: Discriminative, descriptive, and generative models. Quarterly of Applied Mathematics 77.2 (2019): 423-465.

[2] Your classifier is secretly an energy based model and you should treat it like one. arXiv preprint arXiv:1912.03263 (2019).

[3] Should EBMs model the energy or the score?. Energy Based Models Workshop-ICLR 2021.

[4]Learning energy-based models by diffusion recovery likelihood. arXiv preprint arXiv:2012.08125 (2020).

[5] Vaebm: A symbiosis between variational autoencoders and energy-based models. arXiv preprint arXiv:2010.00654 (2020).

[6] A Tale of Two Flows: Cooperative Learning of Langevin Flow and Normalizing Flow Toward Energy-Based Model. ICLR 2022.

[7] VAEBM: A Symbiosis between Variational Autoencoders and Energy-based Models. ICLR 2021

[8] Cooperative training of descriptor and generator networks. PAMI 2018.

[9] Learning Energy-Based Model with Variational Auto-Encoder as Amortized Sampler. AAAI 2021.

[10] Flow Contrastive Estimation of Energy-Based Models. CVPR 2020

**Questions:**

1. What is the advantage of replacing Eq(4) by Eq(5)?
2. In Table 1, what Is the difference between the Central Force Field (Eqn. 4) and the Central Force Field (Eqn. 5)? There looks like no useful information for the Central Force Field (Eqn. 4).
3. Is there duplicated data for the Central Force Field in Table 3 under Unconditional?
4. Could the experiments conducted on a higher resolution, like 64x64? It is hard to see the face details in Figure 5.
5. Are there any other baseline models in section 4.2?
6. How is the stability measured in Table 1?

---

### Official Review · Reviewer_ELK7 · 2023-11-01

**Soundness:** 1 poor
**Presentation:** 2 fair
**Contribution:** 1 poor
**Rating:** 3
**Confidence:** 3

**Summary:**

In this work, the authors proposed to combine both discriminative and generative models inspired by center-force problems in classical mechanics. In doing so, the authors define a new energy for both discrmiatnive and generative models. The authors also conducted experiments comparing hybrid, discriminative and generative  models.

**Strengths:**

It’s always good to draw inspiration from established fields to provide new insight, and this work seems to be good at doing so — the inspiration from classical mechanics is an interesting one, offering fresh perspectives in an unexpected fusion.

**Weaknesses:**

Overall it is evident that the clarity of the manuscript require significant enhancements. While the motivation to connect to classical mechanics is admirable,  there are several concerns regarding the current state of the manuscript.

To begin with, the motivation, although rooted in classical mechanics, seems to be at best only remotely related to the actual proposed frame.  I acknowledge that I may not be familiar with all related literature, yet I find the proposed formulas pretty perplexing, despite reasonable effort to understand:

1. For discrmitative model, the proposed equation 5 seems to diverge form the conventional wisdom regarding the energy-based models in using the exponential of energy, which is in equation 4. The rationale and implications behind this bold decision, which deserves a detailed explanation, remain ambiguous.

2. For generative models, the proposed equation 17 and 18 seem to be overly simplified renderings of score-based models into the central field fashion.  The validity of such simplifications is questionable.

Moreover, the manuscript falls short in dealing with the connection between the proposed framework and the actual training/inference. as well as showing experimental setups. This omission makes it hard to objectively evaluate this work's metric, and constrains readers from gleaning meaningful takeaways. Doing so diminishes the potential impact.

Lastly, the scope and depth of the experiments presented are somewhat restrictive. I do recognize the effort in providing visual results, but I'm afraid that insufficient clarification hurts their demonstrative value.

**Questions:**

N/A

---

### Official Review · Reviewer_mAWA · 2023-11-04

**Soundness:** 2 fair
**Presentation:** 1 poor
**Contribution:** 2 fair
**Rating:** 3
**Confidence:** 4

**Summary:**

This paper proposes a unified foundation model for Artificial General Intelligence that can handle both generative and discriminative tasks. The model incorporates a central force field from physics, which enables the harmonization of energy-based and score-based models. The paper provides a clear explanation of the connections between this method and prior arts, and presents extensive experimental results that demonstrate its effectiveness in both image generation and classification benchmarks.

**Strengths:**

The paper proposes a novel approach to unifying generative and discriminative models using a central force field from physics.

**Weaknesses:**

1. The paper does not provide a comprehensive theoretical analysis of the method. Despite the claimed inspiration from physics, the necessity of introducing a central force field or a compelling motivation is not evident.
2. The paper does not provide a detailed explanation of the model architecture and hyperparameters used in the experiments.
3. The performance of the model is not good.
4. The authors do not discuss the limitations.

**Questions:**

1. Is the table incomplete lacking some results?
2. What is the significance of incorporating a central force field into the proposed method?
3. Since the results (FID) is not promising compared to SOTA, is there any feasible way to improve the performance?
4. Could the authors provide the discussion of the limitation of the method?
5. Could the author provide a more comprehensive and understandable figure to illustrate the method?
6. The theoretical experiments should be conducted to explain the motivation.
7. The paper does not provide a detailed explanation of the underlying physics principles and assumptions that are used to derive the central force field.
8. The paper does not provide a detailed analysis of the computational requirements and resource constraints of the proposed method,